Diel gene expression improves software prediction of cyanobacterial operons

Heller Philip philip.heller@sjsu.edu
Department of Computer Science, San Jose State University , San Jose, CA , United States
Procter James
Electronic publication date: 2022 Apr 11
Publication date: 2022
Volume: 10
Electronic Location ID: e13259
Received 2021 Sep 10; Accepted 2022 Mar 22
Copyright: © 2022 Heller
Copyright year: 2022
Copyright holder: Heller
License: This is an open access article distributed under the terms of the Creative Commons Attribution License, which permits unrestricted use, distribution, reproduction and adaptation in any medium and for any purpose provided that it is properly attributed. For attribution, the original author(s), title, publication source (PeerJ) and either DOI or URL of the article must be cited.
License URL: https://creativecommons.org/licenses/by/4.0/

Keywords: Cyanobacteria, Diel, Operon, Crocosphaera

Funding: Gordon and Betty Moore Marine Investigator University of California, Santa Cruz This work was supported by a Gordon and Betty Moore Marine Investigator grant and the Microbial Environmental Genomics Applications: Modeling, Experimentation, and Remote Sensing (MEGAMER) facility of the University of California, Santa Cruz. There was no additional external funding received for this study. The funders had no role in study design, data collection and analysis, decision to publish, or preparation of the manuscript.

==============================
Cyanobacteria are important participants in global biogeochemical process, but their metabolic processes and genomic functions are incompletely understood. In particular, operon structure, which can provide valuable metabolic and genomic insight, is difficult to determine experimentally, and algorithmic operon predictions probably underestimate actual operon extent. A software method is presented for enhancing current operon predictions by incorporating information from whole-genome time-series expression studies, using a Machine Learning classifier. Results are presented for the marine cyanobacterium Crocosphaera watsonii. A total of 15 operon enhancements are proposed. The source code is publicly available.

Introduction

Photosynthesizing bacteria (phylum Cyanobacteria) are significant participants in global biogeochemical cycles. They arose on earth 3.5 billion years ago (Schopf, 2002), and had oxygenated the atmosphere by 2.5 billion years ago (Tomitani et al., 2006). Cyanobacteria participate in the ocean biological carbon pump (Tréguer et al., 2018), which transports atmospheric greenhouse carbon dioxide to sequestration in the deep ocean. Nitrogen reducing cyanobacteria (diazotrophs) annually convert approximately 200 Tg of atmospheric dinitrogen to bioavailable form (Wang et al., 2019; Tang, Li & Cassar, 2019). Cyanobacteria are used to produce medicines (Tan, 2007; Soni, Trivedi & Madamwar, 2008; Zanchett & Oliveira-Filho, 2013), biofuels (Sakurai et al., 2015; Farrokh et al., 2019), fertilizers (Singh, Khattar & Ahluwalia, 2014; Chittapun et al., 2018), cosmetics (Morone et al., 2019), and food (Khan, Bhadouria & Bisen, 2005).

Despite their ecological and commercial importance, the metabolic processes of many cyanobacteria have not been fully characterized; this is especially true for marine cyanobacteria, which are difficult to cultivate (Zehr, 2011). In particular, identification of operons (consecutive genes on the same DNA strand, controlled by a single promoter and expressed as a single transcript) appears to be incomplete. Operon identification provides clues for the inference of regulatory pathways (Zheng et al., 2002; Westover et al., 2005), supports interpretation of transcriptome experiments (Moreno-Hagelsieb & Collado-Vides, 2002), and can guide annotation of hypothetical genes. The expense of wet lab operon discovery has prompted the development of algorithms for predicting operons from assembled genomes (Zheng et al., 2002; Moreno-Hagelsieb & Collado-Vides, 2002; Price, 2005); predictions from one of these algorithms (Price, 2005) for 1,336 organisms are publicly available (http://www.microbesonline.org/operons/OperonList.html). However, few of these predictions have been experimentally verified and it is possible that operon sizes have been underestimated.

Information for honing in silico operon predictions can be extracted from time-series measurements of gene expression. Many cyanobacterial genes are not expressed at constant rates, but rather exhibit fluctuating transcript abundance in repeating patterns over a 24-h cycle. For example, production of light-harvesting photosystem II proteins, which are only useful during daylight and whose half-lives are generally less than 12 h (Yao et al., 2012; Renger et al., 1989), approximately coincides with available light (Dodd, 2005). Since oxygen disables nitrogenase (the enzyme responsible for nitrogen fixation), diazotrophic cyanobacteria segregate nitrogenase from the oxygen evolved by photosynthesis (Bergersen, 1962; Fay, 1992); segregation is sometimes temporal, with nitrogenase component proteins produced hours out of phase from photosystem II proteins (Tuit, Waterbury & Ravizza, 2004). Diel cycling, defined as a transcript abundance change of at least 2× over 24 h, has been observed in 79% of genes of the diazotrophic cyanobacterium Crocosphaera watsonii (Shi et al., 2010). Since genes in an operon are expected to have similar expression signatures (Sabatti, 2002; Lercher, 2003), a high degree of diel expression similarity among adjacent genes might indicate operon membership. Thus if two predicted operons are adjacent, are on the same DNA strand, and exhibit similar diel expression, then the predicted operons may in fact belong to a single common operon.

The approach presented here uses a machine learning classifier – specifically a logistic model tree (Landwehr, Hall & Frank, 2005; Sumner, Frank & Hall, 2005) (LMT) – to determine when predicted operons in Crocosphaera should be merged. A common metric for quantifying expression similarity is Pearson’s Correlation Coefficient (PCC); however, our earlier work (Heller & Baiju, 2018) has determined that PCC has deficiencies when applied to the current problem. The “Area Between Linear Interpolations of Measurements” (ABLIM) metric, which we have presented elsewhere (Heller & Baiju, 2018), is more appropriate and is the basis of the research reported here. Based on the ABLIM metric, positive and negative example operons were located in the Crocosphaera watsonii genome. A total of 45 kinds of classifier (Table S1) were evaluated, and LMT was selected due to its high accuracy. Adjacent predicted operons were identified as candidates for merging, and the expression similarity of all genes was analyzed by the classifier. A total of 15 pairs of candidate operon predictions are recommended for merging (Table 1).

Table 1 Pairs of predicted operons recommended for merging.

Score	Prior 1	Prior 1 gene functions	Prior 2	Prior 2 gene functions	
0.868	2207
2208	HAD-superfamily hydrolase, subfamily IA, var
No predicted function	2209
2210	No predicted function
No predicted function	
0.834	3078
3079	No predicted function
No predicted function	3080
3081	Glucose-6-phosphate dehydrogenase
OpcA	
0.673	4526
4527	Hydrogenase expression/synthesis, HypA Hydrogenase accessory protein HypB	4528
4529	No predicted function
No predicted function	
0.672	3989
3990	extracellular solute-binding protein, family 3 Amino acid ABC transporter, permease protein	3991
3992	No predicted function
No predicted function	
0.631	5385
5386	No predicted function
Cytochrome c oxidase, subunit II:Cytochrome	5388
5389	Cytochrome-c oxidase
Cytochrome c oxidase, subunit 3	
0.618	1168
1169	3-isopropylmalate dehydratase small subunit
DegT/DnrJ/EryC1/StrS aminotransferase	1165
1166	No predicted function
Glutathione S-transferase, N-term	
0.599	4216
4217	Pentapeptide repeat
No predicted function	4214
4215	No predicted function
HEAT:PBS lyase HEAT-like rpt	
0.576	2640
2641	Phosphopantethiene-protein transferase
ATP-binding region, ATPase-like:Histidin	2638
2639	K+ channel, pore region
K+ channel, pore region	
0.565	6744
6745	No predicted function
No predicted function	6742
6743	Competence-damaged protein:CinA, C-trmnl
Uracil phosphoribosyltransferase	
0.56	4082
4083	Carbamoyltransferase
No predicted function	4084
4085	No predicted function
No predicted function	
0.552	2855
2856	Glycosyl transferase, group 1
No predicted function	2853
2854	Phycobilisome linker polypeptide
Ferredoxin-dependent bilin reductase	
0.537	5116
5117	GTP cyclohydrolase I
Cobalamin synthesis protein/P47K:Cobalami	5114
5115	Cobalamin synthesis protein/P47K:Cobalami
Dihydrouridine synthase TIM-barrel protein yjbN	
0.537	2158
2159	Ribosomal protein L33
Ribosomal protein S18	2160
2161	Exoribonuclease II
No predicted function	
0.518	3464
3465	Porphobilinogen synthase
No predicted function	3466
3467	No predicted function
Metallophosphoesterase	
0.514	3440
3441	Hemolysin-type calcium-binding region
No predicted function	3438
3439	No predicted function
Quinate/Shikimate 5-dehydrogenase	
Note:

Each row presents a consecutive pair of previously predicted operons. The score is generated by a Logistic Model Tree classifier. Each pair presented here has classifier score >0.5, and therefore should likely be merged into a single longer predicted operon. The numbers in the “Prior 1” and “Prior 2” columns are gene identifiers, truncated for formatting; prepend “CwatDRAFT_” to the numbers to generate the full identifier. In the “Gene Functions” columns, genes with unknown function are noted in bold; known function of other genes in a predicted operon can provide clues to the unknown function.

Source code and instructions are available at https://github.com/PhilipHeller/Operons (DOI 10.5281/zenodo.5759925).

Materials and Methods

Computed operon predictions (hereafter the “prior predictions”) for strain Crocosphaera watsonii were downloaded from http://www.microbesonline.org/operons/. Log-expression measurements for 4,407 Crocosphaera genes with eight timepoints were retrieved from a study by Shi et al. (2010). For each gene, log-expressions were normalized to a mean of zero. A positive training set of operons for the classifier was collected by identifying all prior predicted operons in which at least one gene’s expression exhibited diel variation. A negative training set for the classifier was generated by identifying consecutive genes where at least one gene’s expression exhibited diel variation, and where at least one gene is on each DNA strand. All genes of an operon must be on the same strand, to allow correct translation of the transcribed operon.

The classifier requires training and evaluation instances to be represented by vectors of numbers. For each prior in the training sets (and, later, for each merge candidate to be classified), the ABLIM distance between every pair of genes was computed; the instance was represented by a 4-vector consisting of the minimum, mean, standard deviation, and maximum of the ABLIM distances. Gene distances within priors were not considered, as they did not fit a Gaussian distributions and are therefore problematic. A total of 45 classifiers (Table S1) in the WEKA software suite (Hall et al., 2009; Frank, Hall & Witten, 2016) were evaluated on the positive and negative sets using five-fold cross-validation. The Logistic Model Tree (LMT) classifier gave the best accuracy on both the positive and negative data, and was therefore selected for the remainder of the study. The classifier was trained using all the positive and negative instances.

Pairs of prior predictions were identified as candidates for merging (Table 1) if there were no intervening genes, if all genes lay on the same DNA strand and in the same contig, and if each prior contained at least one gene whose expression exhibited diel variation. A 4-vector representation of each candidate was computed as described above, and the representations were evaluated on the trained LMT classifier to generate classification scores (Fig. 1). A candidate was accepted (i.e. all its genes are predicted to be in a single operon) if classifier score was >0.5. Note that this score is not to be interpreted as a probability that the classification is correct.

Figure 1 Classification algorithm.

Prior predicted operons (red, blue) are candidates for merging if their genes are consecutive, if all genes are on the same DNA strand, and if at least one gene in each prior exhibits diel expression.

Software was developed on Eclipse 2020-12 (4.18.0) in Java SE-15 (and is compatible with Java 1.8), using version 3.5 of the WEKA library and version 3.6.1 of the Apache Commons Mathematics Library.

Results

The positive training set consists of the 1,195 operon predictions at http://www.microbesonline.org/operons/. The negative training set is listed in Table S1. A total of 45 classifiers in the WEKA software were evaluated on the training data. The Logistical Model Tree (LMT) had the highest accuracy (Table S1).

A total of 63 pairs of prior operon predictions were identified as candidates for merging. Each prior consisted of two genes, at least one of which exhibited diel expression variation; all genes were on the same DNA strand and in the same contig, and there were no intervening genes between the priors. A total of 15 pairs of priors were classified as belonging to a common operon (Table 1).

Discussion

Diel expression data was combined with prior operon predictions to compute 15 pairs of priors (Table 1) that appear to belong to common operons. It is recommended that each of these pairs be merged into a single prediction.

One reason for honing operon predictions is to gain insight into the function of unknown genes. When unknown genes share an operon with genes of known function, the known function can reasonably be hypothesized to relate to the unknown functions. In Table 1, unknown genes are marked in bold. Six prior predictions include operons where no gene has known function; in all these cases, the present analysis predicts that the prior prediction should be merged with another prior containing at least one gene of known function. Predicted operon membership per se may not be strong enough evidence to infer gene function, but it can provide the basis for hypothesizing function, and the hypothesis can be strengthened by other evidence.

Each operon (training priors and merge candidates) was represented by a 4-vector consisting of the minimum, mean, standard deviation, and maximum of the ABLIM distances among all gene pairs in the operon. None of these statistics alone was sufficient for training an accurate classifier. The LMT classifier had the best accuracy among the 45 classifiers that were evaluated (Table S1). However, this does not imply that LMT should be used when analyzing other organisms. Future work on other organisms should repeat the classifier evaluation reported here, and should choose the best classifier for the organism at hand.

Conclusions

The work presented here demonstrates that machine learning analysis of diel expression studies can improve in silico predictions of operons. When a prior prediction is extended to include genes of unknown function, the function of the known genes in the prior might elucidate the function of the new unknown genes.

The approach presented here can be applied to other cyanobacteria for which diel studies and prior predicted operons are available. Since the method is based on similarity of diel signatures, best results should be expected from organisms whose genes exhibit strong and diverse diel variation. Organisms with weak diel variation can be expected to perform poorly, because the 4-vectors that describe operons to the classifier would all be similar. Experiments with a diel study (Zinser et al., 2009) of the minimal bacterium Prochlorococcus marinus produced poor results with the approach presented here, possibly because the circadian clock mechanism is simplified in Prochlorococcus (Holtzendorff et al., 2008) and its diel genes fluctuate more weakly than those of Crocosphaera.

Supplemental Information

Supplemental Information 1 45 Classifiers evaluated by 5-fold cross validation, in descending order of accuracy.

Classifier names are given as Java class names in the WEKA library. The LMT (Logistic Model Tree) classifier was chosen for this study.

Click here for additional data file.

The author is grateful to Jonathan Zehr, Josh Stuart, Irina Shilova, Rex Malmstrom, and Laurence Nedelec for valuable discussions.

Additional Information and Declarations

Competing Interests

Author Contributions

Data Availability

The author declares that he has no competing interests.

Philip Heller conceived and designed the experiments, performed the experiments, analyzed the data, prepared figures and/or tables, authored or reviewed drafts of the paper, and approved the final draft.

The following information was supplied regarding data availability:

Source code and instructions are available at GitHub: https://github.com/PhilipHeller/Operons (DOI 10.5281/zenodo.5759925).

The third-party data is available from MicrobesOnline Operon Predictions

http://www.microbesonline.org/operons/. Contact Eric Alm (ejalm@mit.edu).

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
