# Peer review of "Diel gene expression improves software prediction of cyanobacterial operons"

_PeerJ, doi:10.7717/peerj.13259_

## Round 0.1 · original submission · Minor Revisions

Two reviewers have examined your work. Both recognise the utility of your approach, but point out a number of relatively minor aspects that need to be addressed.

1. Reproducibility and access to software.
Reviewer 1 (R1) notes that provision of the classifier as a Java Archive (jar) meant they could not access the code. Rather than bunding both documentation and source code as a jar, I recommend you follow PeerJ's software materials guidelines (https://peerj.com/about/policies-and-procedures/#data-materials-sharing) which in brief recommend:
i. uploading source code and documentation to a version-tracked repository (github, gitlab), and
ii. minting a doi for the version used in the publication it via zenodo. OSF or other repositories may also be used here.

To further facilitate access and reproducibility:
iii. upload the working eclipse project to the github/lab repository rather than the exported Jar (this will help those familiar with weka and captures provenance of versions of eclipse used, etc)
iv. translate the word document to a plain text README, and also upload to github.
v. Choose an appropriate license (part of the project creation process).
vi. Specify versions and download locations for the additional libraries (weka, apachecommons). This could be part of the README

2. Both reviewers noted Table 1 seems to have also been uploaded as supplementary info instead of the intended document.

3. R2 notes that P(false+) is not particularly helpful, and remarks that a 'more traditional' statistic may be more appropriate.

4. R2 queries why distance was not used for classification of genes as part of the same operon, and suggests an approach for utilising distance for more precise detection of genes in the same operon.

5. Please also carefully check the abstract and body of the manuscript for minor typos and copy/paste errors.

Reviewer 1 ·

Basic reporting

1. On line 16, please remove "Add your abstract here."
2. The font size and face is different before line 34 and after line 35. Please check that you have the same formatting throughout the paper.
3. Throughout the paper, for e.g. lines 73, 97, 122, Supplemental Table 1 title and legend, I found that you have mistakenly specified "48 classifiers" instead of "44 classifiers". Please check the number of classifiers.
4. Figure 1 seems to be duplicated.
5. In peerj-65502-HellerTable1.docx, the author should correct the number of classifiers used i.e. 44 instead of 48. It also has a typo "Supplemental Table ZZZ2" instead of "Supplemental Table 2".
6. There is no need of peerj-65502-HellerSupplTable2.docx as it only contains "Supplemental Table 2" which is already presented in peerj-65502-HellerTable1.docx.

Overall, I found the paper easy to understand and found no grammatical or spelling errors.

Experimental design

The classifier design was well explained and applied. The proposed algorithm i.e. Logistic Model Tree was specified clearly, and the positive as well as the negative control sets were appropriately chosen. The number of features and timepoints selected to test the classifier seem to be significantly reasonable. Proper normalizations were applied to the gene expression data before classification. Impressively, 44 additional classifiers were tested and the proposed classifier seems to outperform each of them in both negative and positive test datasets.

Additionally, the author was mindful in warning that the classifier may required re-evaluation with other microorganisms and that organisms with little diel variation may perform poorly.

Validity of the findings

The result tables and figures seem to be sound but I could not open peerj-65502-HellerSupplementalCodeAndInstructions.jar on my mac. Does the author have instructions on how to view the contents of the jar file? Alternatively, does the author have the code published elsewhere, for instance Github?

Additional comments

Overall, I found the concept presented by the author simple yet effective in predicting correct operons and assigning functions to unknown genes. I foresee this model being useful to many working with Cyanobacteria and related microorganisms.

Reviewer 2 ·

Basic reporting

Minor grammar comment - in general numbers are written out if they are less than 10 or if they begin a sentence. I do not know if PeerJ adheres to these conventions, but this is what I have seen in most texts.

Ref 31 is posted as being from Zenodo instead of the original journal. It’s okay to put the doi from Zenodo but the journal should be “American Journal of Advanced Research”.

Throughout the text some operons are described as having “both DNA strands present.” This language may be confusing for some readers. I suggest using something like “genes within the same predicted operon are present on opposite strands, so these genes are likely not in the same operon.”

Just a general comment – it looks like Table 1 accidentally got submitted as supplementary material since it was not included as one of the primary files.

Experimental design

Please explain why the distance between genes was not taken into account for a pair of genes to be counted as an operon. There can be genes on the same strand without any intervening genes but they are too far apart to be considered part of the same operon.

Please explain for the positive training set selection why only one gene in the operon needed to express diel variation for the operon to be counted as a true positive. If genes in an operon are expected to have similar expression signatures, shouldn’t this be a requirement for the true positives?

The statistic of P(false +) does not add much to the analysis. I suggest removing this statistic or using a traditional statistic. The reasoning for the cutoff of a 0.51 score is enough.

Validity of the findings

The author checked the predictions by making sure that the genes were on the same strand, same contig, and no intervening genes. However, the distance between the genes also still needs to be checked - even if a pair of genes satisfies these three requirements they still might be too far apart to be on the same operon. One way to find the typical distance between genes in an operon for a specific organism is to take all of the true positives and make a distribution of the distances between genes in the operons. Then the predictions can be evaluated with this distribution.

---

## Round 0.2 · Minor Revisions

Whilst the invited reviewers and I are largely satisfied with your revisions in response to our previous comments, we request the following additional minor revisions:

1. Please address all of R2's requests including those addressing grammar, capitalisation, and inclusion of a statement regarding your observation that the intergenic distances do not conform to any standard empirical distribution.

I also note and appreciate your efforts towards ensuring reproducibility of your work. However, there are still certain critical details that should be addressed:

2. Data availability.

Thank you for uploading the classifier code to github and minting a DOI. However, when I tried to follow the instructions (and methods section) I discovered it is not clear which of the two supplementary tables from Shi et al. was used as input CSV. I can find no correspondence between the column tsv file https://github.com/PhilipHeller/Operons/blob/main/OperonsReleaseFiles/EclipseProjectImage/data/Croco_columns.tsv and the column numbers in either table.

i. Please provide clear instructions detailing exactly how to obtain the data used in the study - and, ideally, also include the exact CSV file used to conduct the analysis (see 4.ii below regarding your right to redistribute derived versions of those published data).

3. Repository/code licensing. The Github project requires a license statement - since Weka is GPL it is most straightforward to select that option for your repository's license. Github has a tool for this: https://docs.github.com/en/repositories/managing-your-repositorys-settings-and-features/customizing-your-repository/licensing-a-repository#applying-a-license-to-a-repository-with-an-existing-license

i. Libraries - weka, commons-maths and opencsv are respectively GPL and Apache 2 licensed codebases. That means you can simply upload your complete lib directory rather than ask others to manually download them. In that case, please remember to provide details of each library's license and its original download location in the Instructions document.

ii. Library versions. It is essential that you state the version of weka, commons-maths and opencsv used for this work. Ideally you should also state what version of Java was used to run the model.

Together, 3.i and 3.ii will allow others to precisely locate the source for each included library, and ensure that your computational set up can be properly reproduced.

4. Data licensing/reuse. Formally, software licenses do not cover data, however there are equivalent permission models that allow data to be shipped along with code.

i. https://www.ncbi.nlm.nih.gov/nuccore/AADV02000001.1/ is freely distributed, and was submitted to genbank to the conditions that the authors place no restriction on distribution and use. For convenience you should consider including it in the data directory (see https://www.ncbi.nlm.nih.gov/home/about/policies/ for details).
ii. Data from Shi et al.. I note that ISME's data reuse policy (https://www.springernature.com/gp/authors/research-data-policy/data-policy-types/12327096) clearly states that data can be freely reused for noncommercial work. It should therefore be sufficient if you clearly cite the origin of the CSV file of time series expression data in the instructions document in the github repository.

Please do not hesitate to get in contact if you would like additional advice regarding my requests.

Reviewer 1 ·

Basic reporting

All proposed changes have been addressed. Thank you

Experimental design

N/A

Validity of the findings

N/A

Additional comments

N/A

Reviewer 2 ·

Basic reporting

Throughout the text, machine learning should be lower case.

Table 1: Do the “?”s mean that a gene function was not able to be classified? Please change to “No predicted function”

Experimental design

Regarding the number of genes in an operon that have diel expression variation for the priors, it looks like in the text it still needs to be changed to reflect this in the methods. Dr. Heller changed the number of priors in the text but did not change lines 85 and 86 to reflect the change to requiring all genes have diel expression variation.

Validity of the findings

I thank Dr. Heller for the effort to do the gene distance analysis of the operons. Please include a sentence in the methods stating that a gene distance analysis of the priors did not fit a gaussian distribution, making gene distance difficult to include in the operon prediction. Even though it did not lead to a contribution to the methods, I think it is worth mentioning the effort and it is informative for the reader.

Additional comments

After the above comments are addressed I can recommend this manuscript for publication.

---

## Round 0.3 · Minor Revisions

Thank you for addressing our requests in the previous round.

Whilst I consider the manuscript ready for publication there remains a problem with the code submitted to the github repository. Unfortunately the instructions do not seem to explain how to correctly run the LMT classifier. They suggest uncommenting the following and editing accordingly:

// To classify merge candidates with an LMT model, writing a report to File otsv:
// Classifier classifier = new LMT();
// exper.evaluateMergeCandidatesWithLMT(classifier, otsv);

However evaluateMergeCandidatesWithLMT is not a defined method for the operons.Experiment class. There is an evaluateMergeCandidates method, but after successful compilation I encountered this runtime error (Java 17 was used):

% java -classpath "src:lib/*" operons.Experiment
??? Cannot invoke "weka.filters.unsupervised.attribute.ReplaceMissingValues.input(weka.core.Instance)" because "this.m_replaceMissing" is null

Whilst I have considerable experience with Java development, I am not an expert with Weka and was unable to locate any documentation that suggested how to correctly configure the missing value behaviour (which is presumably what this exception refers to).

I imagine others would have similar difficulties. Please verify and update the repository with instructions and sample code that allow others.

---

## Round 0.4 · accepted · Accept

Apologies for my less than prompt response. Thanks to your revised project and very clear instructions, I am sure you will be happy to hear that I managed to successfuly reproduce all results (both classifier ranking and output of LMT predictions) under Eclipse 4.21.0 running with AdoptOpenJDK Java 11 and Eclipse Temurin java 17 on OSX Monterey.